# Therapy Development by Genome Editing of Hematopoietic Stem Cells

**DOI:** 10.3390/cells10061492

**Published:** 2021-06-14

**Authors:** Lola Koniali, Carsten W. Lederer, Marina Kleanthous

**Affiliations:** 1Department of Molecular Genetics Thalassemia, The Cyprus Institute of Neurology and Genetics, Nicosia 2371, Cyprus; lolak@cing.ac.cy (L.K.); marinakl@cing.ac.cy (M.K.); 2Cyprus School of Molecular Medicine, Nicosia 2371, Cyprus

**Keywords:** genome editing, hematopoietic stem cell, blood disorders, gene therapy (GT), monogenic disorder, TALEN, CRISPR/Cas, ZFN, base editor, prime editor

## Abstract

Accessibility of hematopoietic stem cells (HSCs) for the manipulation and repopulation of the blood and immune systems has placed them at the forefront of cell and gene therapy development. Recent advances in genome-editing tools, in particular for clustered regularly interspaced short palindromic repeats (CRISPR)/CRISPR-associated protein (Cas) and CRISPR/Cas-derived editing systems, have transformed the gene therapy landscape. Their versatility and the ability to edit genomic sequences and facilitate gene disruption, correction or insertion, have broadened the spectrum of potential gene therapy targets and accelerated the development of potential curative therapies for many rare diseases treatable by transplantation or modification of HSCs. Ongoing developments seek to address efficiency and precision of HSC modification, tolerability of treatment and the distribution and affordability of corresponding therapies. Here, we give an overview of recent progress in the field of HSC genome editing as treatment for inherited disorders and summarize the most significant findings from corresponding preclinical and clinical studies. With emphasis on HSC-based therapies, we also discuss technical hurdles that need to be overcome en route to clinical translation of genome editing and indicate advances that may facilitate routine application beyond the most common disorders.

## 1. Introduction

Hemoglobinopathies, primary immunodeficiencies (PIDs) and congenital cytopenias all share a major commonality: they are hereditary blood disorders caused by genetic aberrations within hematopoietic stem cells (HSCs) that affect production or function of one or more hematopoietic lineages [1]. In principle, each of these diseases can be cured through replacement of the problematic mutant HSCs with genetically normal HSCs, which can then in turn reconstitute a whole new functional hematolymphoid system. Since the first successful transplantation of HSCs to a child with X-linked severe combined immunodeficiency (X-SCID) by Good and colleagues in 1968, allogeneic HSC transplantation (HSCT) has provided the only curative treatment option for patients with such genetic disorders [2]. To date, however, wider application of allogeneic HSCT is restricted by the often unmet requirement of finding a suitable human leukocyte antigen (HLA)-compatible donor and by strenuous pre-transplant myeloablative conditioning regimens, which may cause infertility, secondary malignancies and organ damage [3]. HSCT is further discouraged by frequent short- and long-term side effects of the allografting procedure, such as infections due to immunosuppression regimens, graft rejection and graft-versus-host disease. Under current practices, HSCT is accessible to only 50% of patients for which a suitable HLA-matched donor is available, with a transplant mortality rate of up to 20% [4,5,6].

Owing to these limitations, the concept of autologous HSC gene therapy, in which the patient’s own mutant HSCs are genetically modified, has been gaining momentum, as it offers a potential lifetime cure for a plethora of hematological diseases without the need for an HLA-matched donor and the risk of associated immunological complications. Additionally, the option of reduced-intensity pre-transplant conditioning in an autologous setting allows for faster reconstitution of the hematolymphoid system and contributes to better survival outcomes for patients [7,8]. Initially, autologous HSC-based gene therapy was primarily achieved by gene addition of a functional copy of the disease-causing gene in HSCs via integrating viral or non-viral gene delivery systems, as reviewed elsewhere [9]. Early clinical trials demonstrated safety and efficacy of viral-mediated permanent gene transfer for multiple life-threatening or orphan blood disorders, including congenital hematopoietic disorders: sickle cell disease (SCD) and β-thalassemia [10,11,12,13]; Fanconi anemia (FA) [14]; immunodeficiencies, Wiskott–Aldrich syndrome (WAS) [15,16] and X-SCID [17]; as well as metabolic storage disorders, such as X-linked adrenoleukodystrophy (X-ALD) [18,19] and metachromatic leukodystrophy (MLD) [20]. However, several challenges remained. First, packaging capacity of the most efficient, viral delivery systems imposes restrictions on the therapeutic transgene and hence the diseases that can be targeted via gene addition approaches. Second, for toxic gain-of-function mutations, addition of protein-coding genes alone cannot achieve functional correction. Third, a major shortcoming of permanent gene-addition approaches has been the unpredictability of the integration site of the therapeutic cassette and thus the inherent risk of insertional mutagenesis, oncogene transactivation and aberrant expression of the transgene and neighboring genes [21,22,23,24,25]. Conversely, disease correction by application of conceptually safer, non-integrating vectors remains transient and inefficient by comparison, despite considerable research efforts [26,27]. Taken together, therefore, gene addition has inherent safety and efficiency drawbacks that warrant the search for alternative gene therapy approaches.

Recent advances in programmable nuclease technologies, including zinc finger nucleases (ZFNs), transcription activator-like effector nucleases (TALENs), clustered regularly interspaced short palindromic repeats (CRISPR)/CRISPR-associated protein (Cas) and more recently CRISPR/Cas-based epigenetic, base and prime editing systems have strategically transformed gene therapy approaches. As a major conceptual safety advantage over untargeted gene addition vectors, programmable nucleases and their derivatives share the ability to recognize and bind at specific genomic sites, thus enabling correction of disease-causing mutations or site-specific disruption and integration events without inherent risk of insertional mutagenesis [28]. For many hereditary diseases, physiological and often high-level expression of the affected gene is essential and is readily offered by gene editing tools and their ability to restore normal genomic sequences without the need, common to gene addition approaches, to reduce coding or regulatory sequences. As a safer, albeit currently less efficient, alternative to untargeted gene addition, gene editing tools also allow targeted insertion of entire expression cassettes at so-called ‘safe harbor’ loci, or the precise addition or replacement of functional promoterless sequences at defective gene loci for control under endogenous control elements [29,30]. The latter approach may also address toxic gain-of-function mutations, in contrast to untargeted insertion, while the former approach may also address large deletions or multiple loss-of-function mutations, in contrast to mutation-specific precision repair. Over the last decade, a plethora of pre-clinical studies have provided proof of concept for the therapeutic potential of gene-edited autologous HSCs, vindicated by recent results from clinical trials of gene editing treatment for hemoglobinopathies [31]. Accordingly, many more genetic disorders or disease predispositions that can be targeted through autologous, gene-modified HSCs (see Figure 1) may soon proceed to clinical trials and beyond. Though ongoing studies still uncover as many potential pitfalls for gene editing as they resolve and while many hurdles remain both in the technological field and in the regulatory trajectory, the progress made over the last decade of gene editing technology development has been staggering. Herein, we briefly review key features of the major genome-editing technologies (ZFN, TALENs, CRISPR/Cas9), and provide an up-to-date overview of the landscape of current HSC gene editing therapeutic approaches, highlighting major accomplishments and remaining challenges. Additionally, focusing on findings from corresponding preclinical and clinical studies, we summarize some of the breakthroughs in the field of mutation-specific genome editing for inherited blood disorders. Finally, we critically discuss how current limitations and concerns may be addressed in order to facilitate the translation of these therapies to the clinic.

## 2. The Framework for Gene Editing of HSCs

### 2.1. Overview of Gene Editing Tools

Early attempts for targeted gene editing relied on delivery of donor plasmids with homology arms to the target sequence and on spontaneous action of the cellular homologous recombination machinery to facilitate editing or site-directed integration. Until suitable tools were developed to increase recombination events at the target site, the low efficiency and correction frequencies of this approach were unsuitable for therapeutic application. Following the discovery that double-strand breaks (DSBs) stimulate the action of endogenous DNA repair mechanisms at affected sites, a number of programmable nucleases capable of inducing DSBs at distinct sites were developed for both, gene interrogation and gene therapy. Among the most popular DSB-based editing platforms to date are ZFN, TALENs and RNA-guided CRISPR/Cas nucleases. Nuclease-induced DNA breaks are mainly repaired via two endogenous pathways, (i) the error-prone non-homologous end joining (NHEJ) pathway, which throughout the cell cycle corrects DSBs by ligation of DNA ends but occasionally results in insertions or deletions (indels) at the DSB site, or (ii) the high-fidelity homology-directed repair (HDR) pathway, which is restricted to G2/S phase of the cell cycle and requires the presence of a homologous sequence as a template to repair the DSB [32]. Typically, NHEJ has been used for the disruption, knockout, inversion or tagging of genes, whereas HDR has been used for the precise introduction of desired sequence changes (deletions, substitutions or insertions) at specific genomic sites.

Each nuclease platform is based on a customizable sequence-specific DNA binding domain and a nonspecific DNA cleavage domain. DNA is bound by the eponymous zinc finger and Tal effector nuclease domains for ZNF and TALEN, respectively, and a single-stranded guide RNA (sgRNA) for CRISRP/Cas, and DNA is cleaved by a dimeric FokI nuclease for ZFNs and TALENs and by the monomeric Cas for CRISPR/Cas. Table 1 provides an overview of key features of the most extensively studied gene editing technologies. ZF domains were first discovered in transcription factor IIIa of *Xenopus laevis* and TALE domains in *Xanthomonas* bacteria, with recognition of three and one base pairs per domain, respectively. Modular concatenation of these recognition domains covalently linked to FokI endonuclease allows binding of two ZFN or TALEN monomers to opposite DNA strands either side of the intended cleavage site, which results in target-specific cleavage by the active dimeric form of FokI. On the downside, and while easier to design than older editing platforms [33], both ZFN and TALEN suffer from laborious generation processes due to the requirement for complex protein engineering and cloning protocols [34,35], which made the design of effective nucleases for new targets a bottleneck for the gene editing field. In 2012, derivation of the RNA-guided, sequence-specific CRISPR/Cas9 nuclease system from an antiviral bacterial immune system was set to transform the editing field by removing those limitations. In its engineered form, CRISPR/Cas comprises the Cas9 endonuclease and a single guide RNA (sgRNA) carrying a 20-nucleotide target recognition sequence. With only limitation the additional requirement of a short (typically 3–5 nt) protospacer adjacent motif (PAM) imposed by the type of Cas molecule used, the CRISPR/Cas platform provides a simple, versatile, cheap and predictable platform for gene editing [36,37]. Two key drawbacks of the system are (i) dependence on a cleavage-proximal PAM site, which has been addressed by selection or engineering of Cas molecules with altered or relaxed sequence requirements [29,38,39], and (ii) relatively low on-target specificity and fidelity with correspondingly high off-target activity of the monomeric molecule technology, which has most popularly been addressed by dimeric use of nickase Cas molecules, by employment or engineering of (high-fidelity) Cas molecules with reduced unspecific DNA interaction and by more transient CRISPR/Cas delivery [40]. Towards clinical application, specificity-enhancing strategies are then supplemented with off-target assessment for candidate nucleases, either by saturating evaluation of potential off-target sites or by genome-wide approaches, to allow selection of the highest-specificity tools for downstream application.

Even for highly specific designer nucleases, however, additional concerns remain over the efficacy of precision repair and the general safety of their application. Significant progress has recently been made in raising the efficiency of HDR for therapeutically relevant, DSB-based precision editing even for primitive HSCs [41,42,43,44,45], but concerns still remain over the very employment of DSBs for editing, which may select for apoptosis-deficient cells [41,46,47] or induce inadvertent recombination events [48,49]. Efficiency considerations and acute concerns about the safety of clinical application of DSB-based precision repair thus spawned the search for HDR- and DSB-independent precision editing tools. This came to fruition when David Liu’s group engineered chemical base editors, realized as a nickase-only Cas9 fused to a cytidine deaminase domain to allow permanent C>T/G>A base transition, and later to an adenosine deaminase to allow the reverse A>G/T>C base change [50,51]. Base editing was characterized by high efficiency with minimal indel formation and minimal DSB induction, but also by the risk of inadvertent chemical modification of proximal, by-stander bases and by an inability to create precision indels or transversion (purine-to-pyrimidine and vice versa) events. In another landmark effort to overcome these shortcomings, the same group then combined nickase-only Cas9 with a reverse transcriptase domain and a multifunctional prime editing guide RNA (pegRNA), which serves target recognition and as both primer and template of reverse transcription. While presently less efficient than base editing, the resulting prime editing technology is more versatile and more precise than chemical modification, and allows creation of small precision indels in addition to all 12 possible base-to-base conversions [52]. Besides base and prime editing tools, an arsenal of epigenome editors has been created through modular fusion of catalytically inactive nucleases, such as dead Cas9 (dCas9), with domains of epigenetic modifiers, such as methylases or transactivation domains of transcription factors. Combination of multiple epigenetic modifications can even achieve virtually permanent though reversible transcriptional change that is stable for hundreds of mitoses [53,54].

Thus far, DSB-based and DSB-independent editing technologies have all been demonstrated to correct disease-causing mutations in a number of cell types including HSCs, with an ongoing need to overcome platform-specific shortcomings in safety and efficacy.

### 2.2. Critical Factors for the Clinical Application of HSC Editing

Based on HSCs as editing substrate, the major classes of programmable nucleases have each been exploited as therapeutic tools for the treatment of heritable blood disorders (Figure 2). The strategies employed may be broadly divided into ex vivo or in vivo gene editing approaches and depend on the disease being targeted and the chosen mode of delivery.

Historically, genetic blood disorders affecting hematopoietic-lineage cells, such as hemoglobinopathies, PIDs, and storage disorders, have been addressed by ex vivo manipulation of HSCs, whereas genetic blood disorders affecting extracellular blood proteins, such as coagulation factor deficiencies and hemophilia, have been addressed by in vivo delivery to hepatocytes instead. Only lately have we also seen efforts of HSC-based gene therapy of genetic blood disorders by in vivo delivery, in an effort to reduce treatment-related risks and morbidities. Figure 3 illustrates key steps of in vivo and ex vivo HSC gene editing. Advances in isolation and manipulation of HSCs in combination with well-established HSCT protocols have favored ex vivo modification of HSCs, which has already seen clinical application for several genetic disorders. In spite of these exciting developments, there remain major hurdles that need to be addressed in order for therapies based on gene editing of autologous HSCs to reach their full potential. The factors concerned are: (i) source—the origin and selection of long-term repopulating HSC, (ii) culture—providing the environment for expansion and editing whilst preserving HSC ‘stemness’ and high engraftment potential, (iii) delivery—the application of the editing components to target cell/tissue ex vivo or in vivo and (iv) safety—the avoidance or identification and elimination of recombination events and of off-target cell-entry and editing events, and the prevention of other treatment-related complications.

#### 2.2.1. Source—HSC Origin and Isolation

Constituting less than 0.1% of the adult bone marrow cells, HSCs represent a rare and delicate cell population, which normally resides in specialized microenvironments tethered to the extracellular matrix, osteoblasts and stromal cells that largely determine their behavior of quiescence, self-renewal or differentiation. Collection of HSCs is achieved either via multiple bone marrow aspirations under general or regional anesthesia of the patient or via leukapheresis following enforced egression (mobilization) of HSC from the bone marrow stroma to the peripheral blood. Three Food and Drug Administration (FDA)-approved mobilization agents are currently used in clinical practice: the hematopoietic growth factors, granulocyte colony-stimulating factor (G-CSF) and granulocyte-macrophage colony stimulating factors (GM-CSF), and the small-drug chemokine analogue Plerixafor. A series of clinical trials assessing safety and efficacy of different mobilization agents in the context of HSC-based gene therapy reported that combination of G-CSF and Plerixafor allows enhanced mobilization and sustained in vivo repopulation potential of HSCs [63,64,65,66]. Alternatively, the less efficacious GM-CSF may be used as a salvage strategy for the small number of cases (5–10%) who mobilize poorly in response to first-line regimens [67]. Of note, a major concern associated with collection of HSCs for clinical application is the number of cells that can be obtained for certain diseases such as FA and other bone marrow failure syndromes, where the quantity/quality of HSCs as well as the composition of the bone marrow are significantly compromised. Towards that end, the development of induced pluripotent stem cell (iPSC) technology [68], which provides an alternative source of patient-derived cells, has been exploited by a number of research groups as a renewable source of cells for gene targeting. The potential application of iPSCs for HSC derivation will not be covered further here, but the wide-ranging hopes and concerns surrounding the concept have been reviewed recently elsewhere [69].

Progress has been made in the immunophenotypic definition of primitive, long-term repopulating HSCs (LTR-HSCs), based on surface markers CD34, CD133, CD90, CD38, CD45RA, so that they have variably been defined in human and non-human primates as, e.g., CD34^+^CD38^−^, CD34^+^CD90^+^, CD34^+^CD90^+^CD133^+^ or CD34^+^CD90^+^CD45RA^−^ cells [41,42,43,44,45,70,71]. However, clinical protocols largely rely on single-marker immunomagnetic selection for CD34^+^ [45,72], owing to the empirical long-running success of CD34^+^ cells for curative HSC transplantation, the supportive role of progenitor cells during early post-transplantation myeloid reconstitution, and significant loss of overall yield for LTR-HSCs with highly specific selection protocols. Alas, the CD34^+^ cell population itself is largely heterogeneous, with only a small proportion of cells (1–3%) representing LTR-HSCs [16,73,74], which appear to reside in the CD34^+^CD38^−^ population, while the CD34^+^CD38^0–6%^ population represents virtually all the combined short-term and LTR-HSC potential [75]. However, CD34^-^ cells also contribute significantly to the LTR-HSC pool [76]. A clear-cut positive/negative selection regimen for all LTR-HSCs, as required for comprehensive large-scale cell isolation, is therefore still elusive, but is of critical practical importance. High chimerism and fast reconstitution post-transplantation rely on high cell yield, whereas in particular for gene therapy applications of HSCs, utilization of a purer cell population will help reduce cost and make therapies more widely accessible, as vector requirements are roughly proportional to the number of cells exposed to the vector. In this vein, a number of clinical trials demonstrated successful transplantation of CD34^+^CD90^+^ cells [70,71], whilst several studies showed for lentiviral gene addition that sorting for highly purified CD34^+^CD38^−^ or CD34^+^CD90^+^ HSCs achieved reduced culture volumes, reduced vector requirements, and a significant improvement in transduction efficiency [75,77,78]. Towards clinical application, a good manufacturing practice (GMP)-graded platform based on immunomagnetic bead-based enrichment was developed in the process, allowing enrichment and transduction of CD34^+^CD38^−^ cells, while addition of progenitor CD34^+^CD38^+^ cells post-transduction allowed accelerated myeloid reconstitution [75]. Optimization and adoption of this or other LTR-HSC-selective systems toward specific yet comprehensive capture of LTR-HSCs, reduced vector cost, and minimal processing and culture time would greatly benefit clinical application of HSC-based gene editing.

#### 2.2.2. Culture—Ex Vivo Expansion and Editing of True LTR-HSCs

Efficient genetic modification of HSCs entails ex vivo cultivation and expansion of HSCs. However, removal from their natural supportive microenvironment negatively impacts the delicate HSCs. Moreover, upon collection and immunomagnetic enrichment, the majority of CD34^+^ cells derived from the bone marrow or mobilized peripheral blood are in a quiescent state (G0/G1 phase of the cell cycle). This poses an additional challenge for gene editing strategies that rely on expansion of cells after modification or that are cell-cycle dependent, such as strategies that rely on the, in contrast to NHEJ, mostly S/G2-restricted HDR [79]. Efficient genetic modification is facilitated by the pre-activation of CD34^+^ HSC to exit quiescence, while increased stimulation is also associated with induction of differentiation and impairment of the long-term engraftment capacity of HSC [77,80], which would have dire consequences for clinical application. Such reduced capacity for engraftment and long-term repopulation in particular for HDR-based repair was demonstrated in two exemplary studies by engraftment of modified cells in immunodeficient mice, where long-term (16-week) modification rates for human cells in murine bone marrow resembled the initial rates in transplanted cells much more closely for NHEJ-mediated edits (55%↴46%, 19.8%↴3.3%) than for HDR-mediated edits (11.8%↴2.3%, 17.3%↴0.9%) [81,82]. Currently, conditions optimized for ex vivo culture and stimulation of HSCs, based on serum-free media typically supplemented with recombinant hematopoietic growth factors and cytokines, such as stem cell factor, FMS-like tyrosine kinase 3 ligand, thrombopoietin, interleukin-3 and interleukin-6, result in robust proliferation but also gradual differentiation of HSCs. To overcome this limitation, substantial efforts have been invested into the establishment of new culture and delivery protocols that could increase HDR rates and facilitate ex vivo expansion of HSCs whilst preserving their ‘stemness’ characteristic, as reviewed elsewhere and most recently added to by inhibition of p53-binding protein 1 [60,83,84,85]. Over the last few years, strategies to improve HDR-mediated gene editing via synchronization of cells in S/G2 phases, via induction of HDR components or via pharmacological or genetic inhibition of NHEJ components have been investigated [85,86,87,88,89], including the suitability of different types of HDR donors (see Section 2.2.3 below). In parallel, high-throughput screening studies led to the identification of a number of small molecules, such as prostaglandin E2 (PGE2), StemRegenin 1 (SR1) and the pyrimidoindole derivative UM171, capable of enhancing transduction efficiency in HSCs and facilitating HSC expansion in vitro [77,90,91,92]. However, the effect of these compounds on the long-term repopulating capacity and multilineage differentiation potential of HSCs still remains to be fully investigated.

#### 2.2.3. Delivery—Reaching Target Cells and Sites

Efficient and non-toxic delivery of gene editing tools, which includes the editor and any exogenous homologous donor repair template, into HSCs represents one of the long-standing challenges in the clinical application of gene-edited HSC.

Whereas editors can be delivered in many different forms, HSC gene editing by HDR-based approaches relies on the co-delivery and presence during the editing process of a donor DNA repair template as either an often chemically modified ssODN, a single-stranded DNA (ssDNA) viral genome or, for long targeted insertions, a double-stranded DNA molecule. Utilization of different donors draws on different repair pathways with differing levels of toxicity and editing precision, optimization of which poses challenges in particular for clinical application owing to the recalcitrance and sensitivity of primitive HSCs to the procedure [42,93,94].

For the editors themselves, a variety of non-viral and non-integrating viral delivery systems have been exploited for the delivery of nucleases in vivo or ex vivo over the years, including plasmid DNA, in vitro transcribed RNA, protein for TALEN/ZFN or ribonucleoprotein complexes (RNP) for CRISPR/Cas on as non-viral vectors, and integrase-deficient LV (IDLV), adenovirus (AdV) and adeno-associated virus (AAV) as viral vectors. Each system comes with characteristic advantages and disadvantages. Electroporation with plasmid DNA carrying nuclease and donor template has been the cheapest and most widely used method for editing HSCs. However, toxicity in HSCs combined with concerns over bacterial DNA in the plasmid backbone and an inherent risk of genotoxicity owing to potential random integration of the nuclease expression cassette into the host genome, have restricted its use to proof-of-principle studies [95,96]. Given that duration and concentration of the gene editing machinery within target cells are critical factors determining on-target and off-target DNA cleavage, a ‘hit-and-run’ approach, involving both high and highly transient expression of nucleases within the target cells is usually the main objective. To this end, nuclear transfection or lipofection of in vitro transcribed mRNA or gRNA/mRNA encoding the nuclease and of donor template has become a preferred form of delivery for HSC gene editing, with several studies demonstrating high transfection efficiencies and minimal cytotoxicity associating with this approach [97,98,99]. The transient and robust cytoplasmic expression of in vitro transcribed mRNA minimizes potential risks of off-target insertion or mutagenesis, while several studies have investigated further means of (down-)regulating the editing time window for precision editing [100,101,102]. Conversely, for some applications the short half-life of mRNA may be a limiting factor [103]. Accordingly, several research groups worked towards modifying structural elements of the in vitro transcribed mRNA—notably by incorporation of anti-reverse cap analogues, a poly(A) tail, and 5′- and 3′-UTRs containing regulatory elements—to enhance intracellular stability and translational efficiency [104]. More recently, Wesselhoeft and colleagues reported that circularization of RNA improved RNA stability and was associated with enhanced protein production and stability within the cells [105]. Just like in vitro transcribed mRNA for TALEN and ZFN or mRNA/gRNA for CRISPR/Cas, pre-assembled CRISPR/Cas ribonucleoprotein (RNP) complexes are highly effective vectors for HSC gene editing and allow even more controlled and transient delivery of nuclease activity in vitro [106,107,108]. While RNP electroporation is thus developing into a method of choice for ex vivo modification of HSCs, in vivo delivery for HSCs as an emergent field of interest presents several technical challenges when using purified nucleases. The large size and positive charge of Cas9 protein impedes in vivo intracellular delivery, while Cas9 protein exposure in vivo may additionally induce cellular and humoral immune responses. Indeed, given the microbial origin of CRISPR/Cas, it is not surprising that pre-existing antibodies to Cas9 from *Staphylococcus aureus* (SaCas9) and *Streptococcus pyogenes* (SpCas9) were detected in 79% and 65% of human sera, respectively [109], the effect of which on editing efficiency needs to be minimized. Towards that end, lipid or gold-based nanoparticles, which reduce nuclease degradation and potential immune responses by the host, have been exploited for in vivo gene editing, though the rate of gene editing achieved in HSCs would have been insufficient for most hematopoietic disorders [110,111]. Additionally, a number of non-integrating viral vectors with high tropism toward HSCs, such as IDLVs and AAVs, have been used for delivery of gene editing components. The non-integrating AAV viral vector (and in particular, rAAV6) has emerged as a promising system for delivering nucleases, owing to its high transduction rate in non-dividing cells such as HSCs and its non-pathogenic behavior, which makes it ideal for in vivo and potential therapeutic application. However, its restricted packaging capacity of below 4.8 kb makes it unsuitable for large nucleases such as TALENs and many Cas9 variants. To address this limitation, Bak et al. have recently developed a dual-AAV6 donor vector system (with 6.5 kb packaging capacity) that efficiently targeted T-cells and HSCs with minimum toxicity [112,113]. For their >10 kb cargo capacity, IDLVs and even integrating LVs have popularly been used for the delivery of nucleases and donor DNA templates for research-oriented applications, such as for screening libraries, and IDLVs have also been put forward for clinical application. Both, AAV- and IDLV-derived ssDNA, however, remain in target tissue for extended periods [114,115] and thus poses the risk of ongoing genome modification and of illegitimate integration of the exogenous DNA.

#### 2.2.4. Safety—Avoiding Off-Target Cells and Sites, and Unknowns in Clinical Trials

Quality of gene-edited products can be limited by suboptimal choices for any of the three factors above, source, culture and delivery. Large-scale preclinical studies are required to determine long-term safety, persistence and optimal correction level required to ensure therapeutic benefit from gene editing products for each disorder, while keeping cytotoxicity and potential off-target activity at minimum level.

In contrast to conventional gene therapy approaches, gene editing platforms abolish the need for near-random integration of viral vectors and thus minimize the risk of insertional genotoxicity. However, monitoring of activity and specificity of programmable nucleases remains challenging. One of the biggest concerns associated with the therapeutic potential of gene-edited HSCs is the introduction of unwanted off-target genomic modifications at sites sharing substantial sequence similarity to the intended target [48]. Another concern inherent to DSB-based and even HDR-mediated gene editing is the introduction of on-target indels or recombination events due to inadvertent cellular employment of alternative repair pathways, such as NHEJ-based repair, which may give rise to potentially disruptive indels or in some cases may trigger potentially carcinogenic translocations. Off-target activity has been observed for each one of the four major classes of programmable nucleases [116,117,118,119]. To assess and help minimize off-target activity and associated risks, a growing range of assays have been developed to determine distribution and frequency of off-target events, including in silico tools that are based on sequence homology and computational algorithms, in vitro methods (CIRCLE-seq, Digenome-seq and SITE-seq) and cell-based analytical methods (GUIDE-seq, BLESS/BLISS and HTGTS) (recently reviewed by [120]). GUIDE-seq and Digenome-seq are considered to be the most sensitive genome-wide approaches thus far; however, none of the available methods has proven suitable for use in a clinical trial setting [121]. Methodology for the assessment of chromosomal rearrangements is only now emerging [48,49], with CAST-Seq even allowing quantitative assessment, albeit limited to recombination events between predicted on- and off-target sites [122].

Base and prime editing as DSB-independent editing strategies help reduce or even eliminate the inherent risk of DSB-mediated genotoxic events. Additionally, strategies to reduce off-target activity of editing molecules help reduce inadvertent creation of both, indels and recombination events, also for DSB-dependent editing. Strategies for off-target limitation are published prolifically and, beyond limited exposure to editing tools, include but are not limited to the employment of obligate heterodimeric FokI for ZFN and TALEN, dimeric use of nickase-Cas9 molecules, the establishment of more restrictive PAM sequences, the removal of unspecific DNA interaction in high-fidelity Cas versions, and truncation, extension or bubble-hairpin modification of CRISPR/Cas sgRNAs [40,123,124]

Even as a critical mass of research groups is working on further technology development in all four fields above, progress to preclinical and clinical application of HSC editing has been swift. This has been facilitated in particular with the benefit of preclinical and clinical-trial experience from gene addition approaches concerning source and culture of HSCs, and based in part on efficient ex vivo RNP delivery, combined with saturating off-target analysis to cover the most pressing safety aspect of editing.

## 3. Therapeutic Application of Gene Editing in HSCs

Advances in HSC processing methods in combination with a growing, versatile portfolio of gene editing tools have provided new insights into the underlying molecular mechanisms of diseases and established new frontiers for the treatment of several disorders, so that gene editing has been exploited for a wide variety of therapeutic approaches, including correction of disease-causing point mutations, addition of therapeutic genes to specific genomic sites by targeted integration, removal or disruption of mutant alleles, or regulation of disease modifier expression. Accordingly, the last decade has seen a plethora of studies (Table 2) and 13 clinical trials (Table 3) combining primary hematopoietic human cells and gene editing tools for treatment of non-malignant diseases. 

### 3.1. Hemoglobinopathies—β-Thalassemia and Sickle Cell Disease

The β-hemoglobinopathies, β-thalassemia and SCD, represent the commonest monogenic diseases worldwide, accounting for an estimated 56,000 and 270,000 affected births each year [185]. They are characterized by genetic mutations in *HBB* that reduce or abolish β-globin synthesis in the case of β-thalassemia or lead to formation of a deleterious β-globin variant (*HBB*: c.20A>T, p.E6V in the mature protein) in the case of SCD. Beta-globin is a critical component of adult hemoglobin (HbA, α_2_β_2_), in the absence of which toxic α_4_ homotetramers are formed and insufficient HbA is produced. More than 300 β-thalassemia mutations in *HBB* have been described, associated with imbalanced α-/β-globin chain production, increased apoptosis of progenitor and red blood cells, ineffective erythropoiesis and anemia [186]. Despite improved survival of patients based on conventional treatments, such as life-long blood transfusions and iron chelation for thalassemia and hydroxyurea for SCD, the quality of life of patients is significantly compromised due to transfusion dependency and concomitant complications.

Compared to other disorders treatable by HSCT, hemoglobinopathies as well-characterized and potentially lethal monogenic diseases with a large pool of patients have prompted a disproportionately large number of often pioneering studies on gene therapy development. The first generation of advanced therapeutic medicinal products (ATMPs) was primarily based on lentivirus-mediated addition of a normal *HBB*-like gene copy into HSCs and provided proof of concept for the therapeutic potential of autologous, gene-modified HSCs for β-hemoglobinopathies [10,11,12,13,187]. However, besides common HSCT-related risks owing to myeloablation, insertional mutagenesis through near-random integration is an additional lingering concern. In February 2021, this prompted temporary suspension of trials and marketing for Bluebird Bio’s lentivirus-based Zynteglo, before suspicions over a causative link to one case of acute myeloid leukemia and an initially suspected case of myelodysplastic syndrome were removed [188]. Concerns over insertional mutagenesis have long focused development efforts for novel gene therapies on gene editing, which while still dependent on myeloablation for ex vivo HSC-based strategies, does not inherently pose risks of insertional mutagenesis, and which for DSB-free approaches may even eliminate the risk of inadvertent recombination events [29,30].

Gene editing therapies for β-hemoglobinopathies have primarily been based on three approaches, involving (i) usually HDR-based correction of disease-causing mutations, (ii) addition of an *HBB* transgene to the endogenous locus or (iii) disruption of genes or control elements required for transcriptional repression of the γ-globin genes (*HBG1* and *HBG2*). As to (i), gene correction, the majority of studies aiming to directly correct disease-causing mutations target the SCD mutation (*HBB*: c.20A>T) or the commonest β-thalassemia-causing mutation in East Asia and Southeast Asia (*HBB*: c124-127del) [81,106,107,131,132,133,134,135,136,137,138,142]. For these and others, correction of point mutations most frequently relies on the CRISPR/Cas9 system with ssODNs for HDR-based correction [81,107,132,136,137,138,141], although NHEJ-based editing for mutations outside the open reading frame [152,189,190] and base editing [191,192] have also been employed. Notably, Hoban et al. employed ZFNs for HDR-based editing of the β-globin locus in SCD patient-derived HSCs, where between alternative IDLV and ssODN donor templates, the highest levels of modification were achieved with longer reverse-strand ssODNs of 100 bp [82]. As to (ii), gene addition, in parallel to mutation-specific correction, a number of research groups also evaluated HDR-mediated targeted insertion of the *HBB* transgene to the endogenous locus as a more universal correction approach for restoring normal β-globin levels [106,128,129,139], which given further optimization of HDR efficiencies is a safer, logical next step for therapy by gene addition. As to (iii), gene disruption, a large number of studies has been concerned with the deactivation of factors responsible for γ-globin repression, which in turn leads to expression of the developmentally silenced fetal hemoglobin (HbF), an α_2_γ_2_ heterotetramer that sequesters surplus α-globin, functionally replaces normal adult hemoglobin and has anti-sickling properties. Exploiting the prevalence of NHEJ in HSCs, several groups have focused on reactivation of HbF by NHEJ-based disruption or suppression of silencing factors and regulators of the *HBG1* and *HBG2* genes, including B-cell lymphoma 11A (BCL11A), Krüppel-like factor 1 (KLF1) and ZBTB7A, also known as LRF [193,194]. A critical prerequisite for therapeutic exploitation of *BCL11A* and other genes with multi-lineage expression is the identification of erythroid-specific control elements as targets of disruption, which leave gene function in other cell lineages untouched [143]. A number of studies thus demonstrated reactivation of HbF following nuclease-mediate disruption of erythroid control elements in the *BCL11A* gene or of its binding sites [91,144,145,146,147,149]. Success of these universal approaches in demonstrating highly efficient, safe and precise amelioration of the disease phenotype led to initiation of first human clinical trials evaluating use of CRISPR/Cas9 (NCT03745287) and later ZFNs (NCT03432364) for the treatment of β-hemoglobinopathies (Table 3).

Alternative approaches used by research groups to target β-thalassemia and SCD phenotypes mostly involve emulation of genetic aberrations associated with hereditary persistence of fetal hemoglobin (HPFH), a rare benign condition in which individuals express HbF throughout adulthood and which in β-hemoglobinopathy patients can greatly reduce symptom severity. Traxler et al., by lentiviral CRISPR/Cas delivery and approximate recapitulation of a naturally occurring 13-nt HPFH deletion in the promoters of HBG1 and HBG2 genes, demonstrated amelioration of the sickling phenotype in SCD CD34+ HSCs [144]. Subsequently, Lux et al. delivered TALEN-encoding mRNAs to emulate the same HbF-inducing 13-nt deletion and thereby similarly induced fetal hemoglobin [154]. Drawing on pairs of CRISPR/Cas nucleases, Ye et al. induced a 12.9-kb deletion in the *HBB* locus that left the *HBG1/2* genes intact but largely removed *HBD* and *HBB*, which led to reversal of the SCD phenotype and an increase in HbF levels in SCD CD34^+^ HSCs [145]. Beyond the classical nuclease-mediated gene editing, a number of research groups used less conventional strategies to target β-hemoglobinopathies, highlighting the versatility of therapeutic targets provided by the gene editing technology. In many independent studies that often showcase hemoglobinopathies but would equally be applicable to other diseases, DNA-binding domains of nucleases have been used to modify epigenetic markers or create synthetic transcription or tethering factors [29,30]. Moving from gene editing to transcriptional regulation and epigenome editing in HSCs, the corresponding range of applications is beyond the scope of this article and may merely be exemplified here by a pioneering study by Wilber et al., who fused the activation domain of herpes simplex virus (VP64) to an *HBG1/2*-binding ZF DNA-binding domain to achieve upregulation of HbF to up to 20.9% in primary erythroblasts [195], dependent on the continued presence of the synthetic transcription factor. Finally, just like γ-globin, α-globin is a critical disease modifier for β-thalassemia, so that co-inheritance of α-thalassemia mutations causes lowered α-globin levels and thus ameliorates the β-thalassemia phenotype. This allowed Mettananda and colleagues to restore normal α/β-globin ratios by DSB-mediated α-globin enhancer deletions in edited β-thalassemia CD34^+^ HSCs [148], which unlike γ-globin cannot be therapeutic in its own right, but which would improve treatment outcomes in combination with additional therapeutic agents [196].

Importantly, hemoglobinopathies have also been addressed by DSB-independent gene editing approaches. Liang et al. early on used base-editing to target one of the three most common mutations causing β-thalassemia in China and Southeast Asia (*HBB*: c.28 (A>G)), albeit not in HSC but in a cell system mimicking human embryos [140]. Although precision editing based on base editors still suffers from inadvertent changes to proximal (bystander) off-target bases, the use of a motif-restricted base editor in cell lines has already allowed correction of the *HBB*^−28 (A>G)^ (HBB:c.-78A>G) mutation with minimal bystander edits [192], with implications for future editing of HSCs. Likewise, the highly versatile prime editing technology was employed to revert the SCD mutation (a transversion event: GTG>GAG, V>E), but this has so far only been demonstrated in cell lines [52]. By contrast and though limited to transition edits, which are unable to revert the SCD mutation itself, base editing in CD34^+^ cells was recently employed by two landmark studies with potentially great significance for the future of SCD therapy and for the clinical translation of the therapy of hemoglobinopathies in general. First, targeting the SCD mutation itself and employing a novel base editor with re-engineered PAM requirements, Yen et al. achieved efficient conversion of the sickling variant to the non-sickling variant Makassar (GTG>GCG, V>A) [197]. Second, base editing was used by Zeng et al. to deactivate the BCL11A erythroid enhancer region and achieve therapeutic levels of γ-globin induction in thalassemic CD34^+^ cells [191], an ideal use case for base editors, where bystander edits are not detrimental but might even contribute to inactivation of the target sequence.

### 3.2. Primary Immune Deficiencies

Primary immune deficiencies (PIDs, also known as inborn errors of immunity) represent a heterogeneous group of over 130 rare inherited diseases associated with abnormal immune development and function [198] and globally affecting 1:10,000 live births. In the past years, a number of PIDs, including severe combined immune deficiencies (SCIDs: X-SCID, ADA-SCID), Wiskott–Aldrich syndrome (WAS), chronic granulomatous disease (CGD), and X-linked hyper-IgM (X-HIM) have been effectively treated by conventional gene addition-based approaches. Early clinical trials for X-SCID based on γ-retroviruses, while proving efficiency and safety far beyond conventional treatments, led to leukemia associated with insertional mutagenesis events, in four out of 20 of the pediatric patients [199]. Although an inherent selective advantage of corrected cells in X-SCID may have contributed to the phenomenon, the trial prompted intensified efforts for generally safer designs of gene addition vectors at the time and likewise continues to motivate research into gene editing for a growing list of PIDs as a means of minimizing the risk of insertional mutagenesis [17,43,170,171].

#### 3.2.1. Severe Combined Immunodeficiencies (SCIDs)

SCIDs, characterized by defects in both humoral and cell-mediated immunity, represent the most prevalent (1:50,000–100,000 births) and lethal forms of PIDs. Of these, Artemis (ART)-, *RAG1*-and *RAG2*-SCID show defects in recombination events required for the maturation of T- and B-cell receptors. This has been addressed for ART- and *RAG1*-SCID in human HSPCs by lentiviral transduction with an Artemis-encoding *DCLRE1C* transgene and an RAG1 transgene, respectively, and in associated clinical trials (NCT03538899 and NCT04797260) for lentiviral gene addition [200,201]. By contrast, X-linked SCID, X-SCID or SCID-X1, which accounts for over 40% of SCID cases, has been addressed by several clinical trials based on retroviral gene addition and by substantial preclinical work based on ZFN and CRISPR/Cas designer nucleases. X-SCID is caused by genetic aberrations in the *IL2RG* gene, which encodes the interleukin-2 receptor γ-chain (*IL2Rγ*), a common subunit of cytokine receptors including IL2, IL4, IL7, IL9, IL 15 and IL21 [202]. Defects in cytokine signaling due to lack of IL2RG function result in near-complete absence of T and natural killer (NK) cells as well as impaired development of functional B cells [203]. In the absence of treatment, affected male infants die during the first years of their lives due to inability to fight off bacterial, fungal or viral infections. Early gene therapy trials demonstrated that even low rates of HSC correction could lead to substantial reconstitution of normal T-cell immunity. Gene correction and addition approaches targeting HSCs, iPSCs and T cells have been exploited thus far for the correction of X-SCID-associated mutations and to ameliorate disease phenotype [43,98,155,156,158,159]. A proof-of-concept study led by Sangamo BioSciences, which used ZFNs with a plasmid donor carrying exon 5 of *IL2RG*, achieved functional correction of *IL2RG* gene in K562 and T-cells; though at modest rates [155]. Subsequently, sustained efforts from the San Raffaele Telethon Institute for Gene Therapy to establish ideal culture conditions and timing for the delivery of editing components achieved substantial ZFN-mediated *IL2RG* gene knock-in in patient-derived HSCs, including primitive HSCs [43,156]. More recently, two additional studies demonstrated high HDR-mediated editing for the *IL2RG* locus in X-SCID patient-derived HSCs using ZFNs or CRISPR/Cas9, with the employment of p53 inhibition and AAV6-based donor delivery allowing for up to 40% contribution of corrected cells to grafts in NSG mice [98,159].

#### 3.2.2. Wiskott–Aldrich Syndrome (WAS)

Wiskott–Aldrich syndrome is a rare (1:100,000 births) X-linked recessive PID caused by loss-of-function mutations in the *WAS* gene that encodes for the WASP protein, a key regulator of the actin cytoskeleton and of immunological synapse formation in most hematopoietic lineages [204]. Affected patients typically present with thrombocytopenia, recurrent infections and eczema, and are highly susceptible to developing tumors and autoimmune diseases. Despite improvements in the clinical management, including application of regular platelet transfusions as well as antibiotics, antivirals and antifungals, life expectancy of patients with classic WAS is still limited to a mere 15 years for those with severe forms. Proof of concept for gene editing-based correction of WAS was provided using ZFN-mediated integration of a *WAS* transgene after delivery to patient-derived edited iPSCs [160]. More recently, by combining CRISPR/Cas9 and an AAV6 donor vector, Rai et al. achieved up to 60% targeted integration of therapeutic *WAS* cDNA in patient-derived HSCs, thus setting the foundations for more efficient and safer editing-based therapeutic approaches to treat WAS [161].

#### 3.2.3. Chronic Granulomatous Disease (CGD)

Caused by genetic aberrations in any of the subunits of the phagocyte nicotinamide adenine dinucleotide phosphate oxidase (NADPH) enzyme complex, autosomal recessive (p22^phox^, p40^phox^, p47^phox^, and p67^phox^) or X-linked (gp91^phox^) CGD represents one of the rarest (1:200,000 births) PIDs. CGD patients typically display enhanced susceptibility to severe bacterial and fungal infections and a hyperinflammatory state owing to an inability to resolve corresponding inflammations. Treatment modalities include lifelong antibiotic/antimycotic prophylaxis as well as immunomodulation with interferon-γ. X-linked CGD, which accounts for over 60% of CGD cases, has been the target of all clinical CGD trials thus far. Likewise, autosomal recessive CGD has to date only been addressed by editing based on iPSCs and cell lines [168,169,205], while several studies for X-linked CGD have already drawn on HSCs. In a 2016 study, ZFNs and AAV6-based donor templates achieved target integration of a *CYBB* transgene, encoding the gp91^phox^ protein, into the inert (‘safe harbor’) AAVS1 site and demonstrated engraftment of cells in the bone marrow of NSG mice 17 weeks post-transplantation [97]. In further studies employing CRISPR/Cas9 in combination with ssODN, the same group initially reported 12–31% targeted correction of the *CYBB* 676C>T point mutation in X-CGD patient-derived HSCs [167], before transient inhibition of p53-mediated apoptosis and transplantation into immunodeficient mice enabled up to 80% the level of gp91^phox^-positive long-term repopulating cells seen with healthy donor cells as positive controls [89].

#### 3.2.4. Immunodysregulation Polyendocrinopathy Enteropathy X-Linked (IPEX)

IPEX is a rare X-linked disorder, with under 300 known cases globally, characterized by T cell defects and autoimmunity, and brought about by mutations in the *FOXP3* gene, a master regulator of regulatory T (Treg) cells [206,207]. Although *FOXP3* gene addition to CD4^+^ T cells with constitutive expression has as a therapeutic effect, life-long correction across multiple sub-lineages would require physiological expression and employment of HSCs instead [208,209]. To this end, HSC-based repair was recently performed by CRISPR/Cas-mediated site-specific integration into the first coding exon of the *FOXP3* endogene, using a 4.4-kb HDR donor for AAV6-based delivery [170]. In addition to the 1.3-kb FOXP3 cDNA, the donor included the truncated neuronal growth factor receptor (tNGFR) as a clinically compatible selectable marker, allowing enrichment to 29 ± 8% tNGFR-positive cells and long-term reconstitution of the immune system in NSG-derived mice. Reconstitution was low, which may in part have been attributable to inclusion of the 1.6-kb tNGRF expression cassette in the donor construct [106].

#### 3.2.5. Hyper IgM Syndrome (HIGM)

Hyper IgM (HIGM) syndrome is characterized by defective B cell hypermutation and recombination, which prevents immunoglobulin class switching from IgM to other immunoglobulin isotypes and thus leads to susceptibility to microbial infections [210]. As a group of disorders, HIGM types 1 through 5 are variably brought about by mutations in genes affecting DNA deamination (*AICDA*), DNA cleavage (*UNG*) and CD40 signaling (*CD40*, *CD40LG*), of which the latter is required to induce class switching by B cell interaction with T cells. The vast majority of HIGM patients are male and affected by X-linked hyper-IgM syndrome (HIGM1), caused by mutations affecting T cell expression of the CD40 ligand CD40LG. For HIGM as for IPEX, correction of T cells demonstrably provides short-term therapeutic benefit [171], but curative treatment would once more have to rely on correction at the level of HSCs for reasons of cell longevity and because specifically for CD40 and CD40LG, expression is also relevant in other cell types, such as myeloid cells [211]. Additionally, highly regulated transient expression of CD40LG appears to be required in order to avoid malignancies [212]. Addressing all of these concerns for HIGM1, Kuo et al. employed TALEN and CRISPR/Cas9 nucleases for targeted insertion of a CD40LG cDNA downstream of the endogenous CD40LG promoter and upstream of any known mutations causative of HIGM1 [99]. In the event, AAV6-based delivery of the HDR donor was by far superior to IDLV-based delivery, and colony-forming assays and bulk analyses of HSC-derived cells showed up to 28% gene modification rates for both nuclease platforms. In in vivo studies, transplantation resulted in 60% NSG mice showing thymic reconstitution and in 80% long-term-positive NSG mice with on average 4.4% integration rate. Given the dynamic nature of CD40LG/CD40 interaction, even 10% of correction in peripheral cells might facilitate class switching at therapeutic levels, so that minor improvements in methodology might be sufficient to reach clinically relevant efficacy in HIGM1 patients.

### 3.3. Congenital Cytopenia/Inherited Bone Marrow Failure Syndromes

Congenital cytopenias, the inborn absence or reduction of one or multiple HSC-derived cell lineages, are caused by inherited or spontaneous germline mutations affecting the bone marrow [213]. The most common of these rare anemias are Fanconi anemia (FA), Shwachman–Diamond syndrome, Blackfan–Diamond anemia, dyskeratosis congenita (DC), congenital amegakaryocytic thrombocytopenia (CAMT) and reticular dysgenesis. In most cases and in particular with early treatment, HSCT substantially prolongs survival, although syndromic features outside the HSC lineages, such as early onset squamous cell carcinoma in FA and DC, are not addressed. The same limitations apply for the therapeutic use of corrected autologous HSCs, where HSCs affected by different congenital cytopenias, including FA, have the additional disadvantage of being inferior substrates for correction. However, this may be compensated by an apparent in vivo proliferation and survival advantage for corrected cells [174]. To date, gene editing in the context of most congenital cytopenias has been restricted to experimentation in cell lines or model systems [30,214], with the notable exception of substantial efforts and successes for FA (as summarized below) and of the proof of principle for mutation-specific CRISPR/Cas-based repair in CD34^+^ cells for CAMT [181].

#### Fanconi Anemia

FA is a rare (1:100,000) hereditary disorder characterized by mutations in one or more of the 22 described *FANC* genes that result in defects in DNA damage response, physical anomalies, and progressive bone marrow cell underproduction. Affected patients typically present with bone marrow failure, physical abnormalities and organ defects and are particularly prone to developing cancer. Early clinical trials emphasized the challenges associated with the collection and manipulation of autologous HSC from affected patients, but at the same time demonstrated the strong engraftment capacity and proliferative advantage of corrected HSCs over mutant HSCs from FA patients [215]. Thus far, a number of research groups have exploited use of nuclease-mediated gene editing as an alternative to the use of LV-based gene addition approaches [173,174,175,216]. Owing to the extensive damage in the bone marrow niche of FA patients and poor quantity and quality of HSCs, early proof-of-concept studies used FA-patient-derived fibroblasts as a model for targeting *FANC* mutations using the CRISPR/Cas9 system [173,216]. However, Diez et al. in addition to FA-patient-derived cell lines and healthy CD34^+^ cells also used FA CD34^+^ cells to demonstrate the feasibility of correcting the FA phenotype by ZFN-mediated introduction of an *FANCA* donor into the AAVS1 ‘safe harbor’ locus [174]. A key obstacle in the efficient use of HDR for targeting causative mutations in FA was recently highlighted by the Corn group, who demonstrated implication of FANC proteins in Cas9-mediated single-strand template repair [217] and in the utilization of double-stranded donor DNA for HDR-based repair [218]. Notably, Román-Rodríguez bypassed this difficulty by employing NHEJ-mediated disruption in order to introduce compensatory mutations instead, which allowed restoration of expression from the FANCA, -C, -D1 and -D2 genes [175]. Alternatively, small-molecule inhibitors of HDR-based repair may be employed to enhance HDR efficiencies, even in an FA background [218].

### 3.4. Inherited Bleeding and Clotting Disorders

Bleeding and clotting disorders comprise conditions with excessively long bleeding (hemophilia) and others with excessively fast blood clotting (thrombophilia/hypercoagulability), which in their extreme forms may be lethal. Both disorders inversely deregulate a complex blood enzyme cascade that normally stays inactive in absence of injury and that effectively seals the walls of blood vessels by clot formation when damage to the endothelium and subendothelium is detected. Hemophilia and thrombophilia mutations are mutual disease modifiers, so that co-inheritance of thrombophilia risk factors leads to milder phenotypes for hemophilia [219]. For thrombophilia, causative mutations include those in factor V Leiden and the 2021G>A mutation in prothrombin, and, in rarer but generally more severe cases, mutations in antithrombin III, protein C and protein S [220]. For hemophilia, a total of over 1000 mutations in the X-chromosomally encoded clotting factors VIII and IX are responsible for hemophilias A and B, respectively, and mutations in factor XI on chromosome 4 for hemophilia C [221]. From among the bleeding and clotting disorders, all advanced preclinical and clinical gene therapy studies to date have focused on hemophilia A or B.

#### Hemophilia

Natural production of coagulation factors VIII and IX occurs in the liver, so that most published gene therapy approaches employ AAV vectors for in vivo gene addition or gene editing in hepatocytes [179,222,223,224]. Of particular note here is the principle of cDNA expression for factors VIII and IX under control of the endogenous albumin gene promoter, after ZFN-mediated targeted integration [225]. However, several new approaches employ ex vivo manipulation of HSCs for therapy development instead [226]. For instance, clinical trials for HSC-based lentiviral gene addition approaches include as strategies the systemic expression of factors VIII or IX [227,228] and the expression of a platelet-retained factor VIII in an attempt to avoid unnecessary exposure of factor VIII to the immune system [229]. Editing approaches now draw on the same principle of HSC-derived expression, including targeted integration of a factor IX cDNA under control of the endogenous α-globin promoters [162], where high levels of expression for the dispersible factor may compensate for low integration efficiencies and correspondingly small chimerism of edited long-term repopulating HSCs.

### 3.5. Beyond Blood

Beyond application to blood disorders, HSCT may also serve to replace phagocytic cells acting in solid organs in hemophagocytic conditions, or to provide lifelong enzyme or other protein therapy in metabolic diseases or protein deficiencies [3]. For all of these conditions, based on the modification of autologous HSCs and even beyond HSC-derived tissues, gene therapy and gene editing might therefore once again allow the same therapeutic benefits as conventional HSCT but without the risks associated with allogeneic transplantations. The vast majority of studies and successes for HSC-based protein delivery outside the blood circulation have been noted for congenital metabolic disorders, but new targets are emerging.

#### 3.5.1. Inborn Errors of Metabolism (IEMs)

IEMs describe a large heterogeneous group of genetic diseases that generally result from a defect in an enzyme or transport protein and a corresponding block in a common metabolic pathway. This leads to toxic accumulation of macromolecules and to cellular dysfunction. IEMs are individually rare, but as a group have a frequency of up to 1:800, depending on the population. Phenylketonuria (PKU) and medium-chain acyl-CoA dehydrogenase (MCAD) deficiency with respective incidences of 1:10,000 and 1:20,000 are among the most prevalent [230]. Also among the more common IEMs, lysosomal storage diseases are a group of about 50 rare inherited metabolic disorders that result from defects in lysosomal function. Other major classes of IEMs are glycogen storage disorders and peroxisomal disorders [231]. Palliative treatments, e.g., by enzyme replacement therapy, is costly and not universally effective. Similar to hemophilia, metabolic disorders may be treated by in vivo editing of hepatocytes and enzyme expression from the endogenous albumin promoter, as has been demonstrated in mice for proteins deficient in Fabry disease, Gaucher disease and Hurler and Hunter syndromes [225]. Importantly, for many of these disorders, HSCT of enzyme-rich donor cells has been performed as therapy for over 20 years, with varying success depending on the specific enzyme deficiency and the stage of the disease [232]. This prompted Pavani et al. to perform targeted integration in HSCs to address enzyme deficiencies underlying Fabry disease, Wolman disease and Hurler syndrome, in line with their work on α-globin-promoter-driven expression of clotting factors for hemophilia in the same study [162]. As for bleeding disorders, the potential for high level expression and the easy accessibility of HSCs for manipulation might give HSC-derived expression of soluble factors enormous importance for future curative therapies of metabolic and other disorders.

#### 3.5.2. Neuropathies

HSC-derived cells may pass the blood–brain barrier and thus facilitate engineered interactions or protein secretion with therapeutic effect. This is exemplified by the delivery of pro-apoptotic compounds to achieve reduction of brain tumors in mice [233]. A notable recent development for this strategy based on gene editing technology is application to Friedreich’s ataxia, a neurodegenerative disease caused by pathogenic GAA trinucleotide repeat expansion in the *FXN* gene. Transplantation of normal HSCs into a mouse model of the disorder allows transfer of the normal protein from HSC-derived immune cells to affected neurons and myocytes, with therapeutic effect [234]. Towards clinical translation of the same principle by transplantation of corrected HSCs, CRISPR/Cas-mediated excision of the GAA expansion in human HSCs allowed normal mitochondrial function and hematopoietic differentiation of cells [235].

### 3.6. Acquired and Complex Diseases

Therapeutic application and thus disease examples in this review are focused on application of HSC editing for the treatment of inherited and in particular monogenic diseases. It is of note, however, that gene editing of HSC and corresponding lineages has also been applied to combat acquired and complex diseases, including infections and cancers, as has recently been reviewed elsewhere [236,237,238,239]. A prominent example for the former is engineered resistances against HIV-linked acquired immunodeficiency syndrome (HIV/AIDS), where CD4 co-receptor CCR5 and chemokine receptor CXCR4 mediate infection of different HIV isolates and may be edited or silenced in T cells or HSCs to make HIV target cells resistant to the virus [182,183,184,240]. The most prominent application of gene editing to cancers, with wide-ranging clinical implications, is in the creation of chimeric antigen receptor T and NK cells for effective immunotherapies against neoplasia [239]. Beyond application to therapy, gene editing in HSCs is already contributing to genome-wide genetic screens, to a growing understanding of pathways and driver genes underlying neoplasms and other complex or acquired conditions for which HSCT is currently employed as a treatment modality [241,242,243,244]. As escape of individual cells would be detrimental in neoplastic conditions, it is particularly for the treatment of non-neoplastic complex conditions affecting HSCs and their offspring, that gene editing might have a direct therapeutic role in the future. Many of these disorders have a genetic component or feature characteristic changes in cellular pathways as tentative pointers towards therapeutic targets [245,246].

## 4. Outstanding Challenges En Route to Clinical Application

Building upon more than 50 years of HSCT experience and on knowledge from a myriad preclinical trials and over 120 clinical trials involving HSC-based gene therapy interventions by gene addition, gene editing therapies using HSCs have moved from bench to bedside in less than a decade. Besides ongoing technical innovations in cell isolation and culture, gene addition trials and pioneering commercialization of corresponding products have sped up the clinical translation of gene editing by establishing corresponding business, manufacturing, clinical and supply infrastructures, and by initiating the international harmonization of corresponding regulatory bodies and guidelines [84]. Specifically for gene editing, existing applications already demonstrate a high level of versatility and a potential for safety and efficacy beyond what can be achieved with gene addition. However, many challenges and drawbacks remain.

### 4.1. Technical Challenges and Practical Limitations

Efficient HSC intracellular delivery, fine-tuned editor activity and high specificity of programmable editors are prerequisites for successful clinical translation. Beyond the editing technology itself, parameters that need consideration for clinical trial protocols include HSC source and quality, the patient’s health status, disease and modifier genotypes, and the choice of conditioning regimen. Of all these factors, choice of cell material, appropriate culture, precision of editing and avoiding myeloablation such as by in vivo procedures are important targets for optimization.

#### 4.1.1. Cell Yield and Composition

With current protocols, obtaining a sufficient number of HSCs for efficient ex vivo genetic modification and subsequent transplantation may not always be possible, given that certain diseases affect the bone marrow and diminish HSC quantity and quality. This is aggravated by the delicate nature of primitive HSCs and possible cell losses by enrichment, culture, freezing and thawing procedures during the manufacturing process [247]. Enrichment of a more HSC-specific subcompartment of CD34^+^ cells in order to improve LTR-HSC manipulation and lower vector requirements therefore needs to be balanced against a corresponding loss in cell yield. Moreover, infusion of a mixed population of gene-edited HSCs and unmanipulated hematopoietic progenitors HSCs, which may have been spared ex vivo culture and genetic manipulation, was reported to improve engraftment and hematopoietic reconstitution [75,77]. Hence, in addition to the requirement by the majority of clinical trial protocols for a small number (1.5–2.0 × 10^6^ CD34^+^/kg) of unmanipulated cells to be stored as ‘back-up’ in case of engraftment failure for gene-modified HSCs, preservation and possible progenitor selection of additional un-manipulated cells for co-administration with edited cells may improve myeloid reconstitution following autologous HSCT.

#### 4.1.2. Maintaining Stemness

Cultivation of HSCs is required for efficient ex vivo editing of HSCs but has been shown to negatively impact their stemness characteristics and engraftment capacity. To overcome this issue, a number of stem cell niche modulants (PGE2, SR1, and UM171) have been introduced in culture protocols to maintain and expand the most primitive long-term repopulating HSC fractions that contribute to the graft. Further clinical safety and feasibility studies including these modulants will allow assessment of their ability to ensure preservation of the long-term multilineage potential of gene-edited HSCs without side effects [248,249].

#### 4.1.3. Precision Repair

The predominance of error-prone NHEJ-mediated repair and limited efficiency of HDR in the most primitive HSCs has been a long-standing challenge for HSC gene editing therapies, as DSB-independent precision editing by base editing already shows it potential for clinical translation in HSC precision repair [191,197] and as prime editing is beginning to achieve high-efficiency precision edits in primary cells [250]. Towards improving DSB-based precision editing of HSCs, many studies have endeavored to inhibit NHEJ, boost HDR or regulate nuclease activity, while optimal conditions to extended activation time and allow establishment of HDR-enhancing conditions for HSCs without impairing their functional characteristics are still elusive [86,88,251]. Additionally, removal of unedited primitive stem cells, which could potentially compete with edited cells for engraftment, may provide an alternative route to compensate for insufficient editing efficiency and HDR frequency [106,113], although inclusion of selectable markers to do so may come with its own drawbacks [106,170].

#### 4.1.4. Towards Safer Conditioning and In Vivo Editing

Ex vivo HSC editing has been associated with adverse events in clinical trials, and in particular patient conditioning has been a target for improvement in several studies. Full myeloablation with cytotoxic alkylating agents such as busulfan is associated with elevated treatment-related mortality in some conditions, such as X-SCID, where milder conditioning with a mixture of agents was found to be sufficient to allow reconstitution of the B cell compartment with donor cells instead [252,253,254]. The same is also true for other PIDs, where alternative conditioning has to be evaluated on an individual basis [254], and for hemoglobinopathies [255,256]. For other disorders, such as FA, efficient therapy is possible without conditioning altogether [14], while innovative, selective conditioning methods, such as cell-type-specific drug delivery targeting CD117 or CD300f, and the use of non-genotoxic agents, such as saporin, are being investigated for a range of disorders to minimize the risk of treatment-related hematopoietic malignancies [55,257,258,259].

Importantly, in vivo gene editing of HSCs using virus-based or lipid-based nanoparticles may avoid conditioning altogether and could bring about a breakthrough in HSC gene editing therapies by providing a potentially safer and more readily applied therapeutic regimen. However, this has so far only been applied in mouse models [59,62,150], and scaling up the procedure will pose new challenges. Benefits of this approach include not only its independence from myeloablation and cell isolation procedures, but also avoidance of the logistics surrounding patient-specific modified cells as the therapeutic product, while its substantial current drawbacks include the risk of sub-therapeutic efficiencies and of editing off-target and even germline cells. Of note, the majority of disorders treatable by HSCT are recessive and often associating with diminished protein quantity or function. Given frequent functional redundancy in biological systems and in vivo selection of corrected cells, correction of only a proportion of diseased cells is thus often enough to reverse disease pathology and ameliorate symptoms. Indeed, post-transplant follow-up studies indicated mixed hematopoietic chimerism as low as 10–30% to associate with amelioration of clinical symptoms and transfusion independency in patients with SCD, thalassemia, SCID and other PIDs [260,261].

### 4.2. Safety

Gene editing-based therapy represents a relatively young research field and despite already remarkable accomplishments, data on long-term risks, persistence of editing effects and on safety of gene-edited HSCs are still missing. One of the key lessons learnt from decade-long clinical application of initially γ-retroviral and then lentiviral vectors for gene addition is an appreciation of the residual risks and of lingering unknowns with every new ATMP technology, even in the hands of the most conscientious and knowledgeable actors [262]. Moreover, clonal analyses possible during the follow-up of clinical gene addition trials revealed stages of clonal HSC latency and contribution to blood reconstitution [73,263], which will also determine the efficacy and long-term safety of gene editing trials. Looking forward and in close correlation to risks encountered for gene addition, major concerns for gene editing technology presently include the dangers of DSB induction, long-term stability of edited cells, immunogenicity of treatment, and the overall proportionality of risk vs. benefit.

#### 4.2.1. DSB Induction

Nuclease-mediated gene editing could trigger intracellular defense mechanisms and downstream responses affecting proliferation, viability and differentiation of HSCs [264]. In particular, safety concerns arise due to the inherent dependence of traditional nuclease editors on efficient DSB induction, which may induce a p53-mediated DNA damage response, cell cycle arrest or alterations in HSC stemness [41,46,265]. Several studies have demonstrated that transient inhibition of p53 decreases the DNA damage response and increases HDR levels [89,98,159], but further work is required to evaluate the safety of using such inhibitors, given the tumor suppressive function of p53, in a setting where even target-specific DSB-dependent editing events contribute to hematopoietic abnormalities [46,47,48,49,266]. As single on-target DSB event may already prompt chromosomal rearrangements with potentially catastrophic consequences, inadvertent cleavage at sequence-similar off-target sites will exacerbate the occurrence of recombination events or might induce indel formation, which could lead to adverse events by transactivation of oncogenes or inactivation of tumor suppressor genes. Similar to the risk of insertional mutagenesis for gene addition, therefore, the risk of off-target mutagenesis in HSCs for gene editing raises long-term safety concerns owing to their unique capacity of both self-renewal and pluripotency. Notably, the nickase-mediated base and prime editing tools may display sequence-dependent off-target activity just like DSB-based editors, but they have a substantially lower tendency to induce DSBs and indels at on- and off-target sites. Wherever possible, DSB-based editing is therefore increasingly replaced with base editing or prime editing technologies, although the latter is still subject to substantial optimization [267].

#### 4.2.2. Long-Term Stability

Achieving sustained and therapeutically relevant levels of correction in the final gene-edited product is one of the most critical aspects in curative treatments. In spite of reportedly good engraftment rates of gene-edited products achieved in the majority of preclinical studies, recent studies indicate a decrease in frequency of edited cells within 8–16 weeks following transplantation [81,87,106]. Whether this is down to inefficient gene editing in primitive HSCs or to their inability to self-renew after ex vivo culture and manipulation remains to be determined. Currently, the majority of clinical studies based on HSC gene editing are still at an early stage, which combined with a usually small sample size impairs general predictions about long-term behavior of edited HSCs in vivo.

#### 4.2.3. Immunogenicity

In vivo delivery of designer nucleases poses bigger hurdles than ex vivo application by the immunogenicity associated with the introduction of foreign particles into the human body, creating the need for patients to undergo immunosuppression therapy. This is of particular concern for in vivo gene editing with the CRISPR/Cas platform, because of frequently pre-existing immunity to the protein component [109]. Besides the risk for accumulating off-target mutagenesis, the potential for immunogenicity with prolonged exposure to Cas9 is therefore another argument for highly transient expression of editing components. Other ways of addressing this challenge for gene editing in the future may be conventional immunosuppression regiments [268], the ex vivo expansion and reinfusion of autologous regulatory T cells before treatment [269] or, more fundamentally, targeted epitope elimination [270] for Cas9 or therapeutic use of Cas9 variants for which pre-existing exposure is unlikely.

#### 4.2.4. Proportionality

Akin to the universal setback the field of gene therapy experienced following leukemia cases in the early X-SCID gene addition trials, any adverse events for therapies based on HSC gene editing could jeopardize societal support for the entire class of treatments. Moreover, advances in supportive care and improvements in quality of life and survival of many patients with various hereditary disorders call for a careful assessment of potential risks and benefits of therapies by gene editing compared to conventional treatments. For patients with invariably fatal inherited disorders or not responding to conventional treatment, the still risky strategy of gene editing would seem acceptable, for other cases it would not. Even in the absence of adverse events, irresponsible use of editing technology is unacceptable and will rightly draw societal backlash, as observed for a wholly gratuitous and technically deficient use of germline editing to engineer prenatal resistance to HIV in 2018 [271].

### 4.3. Cost and Public Access 

Over the last decade, the ATMP product market has substantially expanded and is expected to be worth over USD 11.2 billion by 2025 [272]. However, widespread application is impeded by the high cost of ATMPs, which is in part based on a shrewd assessment of the patient’s and health provider’s willingness to pay, but also based on high development and production cost, high safety standards, performance-related reimbursement models and a current shortage of clinical-grade reagents. Despite 14 ATMPs gaining market authorization between 2009 and March 2019, four have already been withdrawn for commercial reasons, and the gene-addition β-hemoglobinopathy drug Zynteglo has just been withdrawn from Germany, Europe’s biggest individual economy [273], over pricing disagreements. These events indicate that in the nascent business of one-off curative ATMPs based on HSCs, key remaining challenges are access to therapy and the affordability and commercial sustainability of products and services.

#### 4.3.1. Access

Despite increased interest by investors and the public, a major challenge faced by gene-therapy-based ATMPs following clinical evaluation and approval are cost and patient accessibility. A major factor driving the high cost is the large amount of clinical (good-manufacturing practice, GMP)-grade products that are required for manufacturing GMP-grade edited HSCs. In contrast to research-scale cell editing, which is typically performed with 0.2–1 × 10^6^ CD34^+^ cells, the target dosage for clinical-scale cell editing and transplantation is 2–20 × 10^6^ CD34^+^ cells/kg, which consumes an enormous amount of costly GMP-graded reagents and assays required for product development and quality accreditation. Additionally, for several disorders, assessment of ex vivo gene editing may be required for multiple hematopoietic cell lineages, which further contributes to the length and cost of the manufacturing process. Moreover, only a small number of medical centers around the globe have clinical-level expertise in gene therapy and gene editing treatments, which shortens supply, necessitates cell and reagent transfers and further restricts access to treatment for patients. An increasing partnership between pharmaceutical companies and academia helps accelerate clinical translation of gene editing technologies, while likely stifling the development of ATMPs for rarer and likely unprofitable diseases with small patient cohorts. Additionally, many patients affected by hereditary hematological disorders live in developing countries, such as SCD patients in Africa and many thalassemic patients in the Middle East and Southeast Asia, while business and research for ATMPs take place elsewhere. The geographic distribution of stakeholders in ATMPs combined with pressure on the companies involved to provide returns on investment raises the questions whether there will ever be a drive to making emerging and lifesaving treatments accessible and affordable to the majority of patients.

#### 4.3.2. Sustainability and Affordability

In order for autologous HSC gene editing-based therapies to meet their potential, development of new financial models and reimbursement policies and processes are required to enable patient access to these treatments while allowing companies to thrive. Regrettably, ATMPs are often considered as products with little commercial value and high commercial risk given the often-small patient cohort size and the lengthy and complex manufacturing process that is tailored to individual patient. The majority of disorders treatable by editing of HSCs are moreover in their majority caused by numerous mutations across multiple genes, which in the absence of universal disease modifiers multiplies the ATMP development effort and investment required to bring treatments to the market and to patients. Cost might be reduced by in vivo HSC gene editing, which would conceptually provide a more elegant system of manufacturing and supply, by drawing on off-the-shelf products that could be administered directly to patients. For further progress here, major hurdles associated with target tissue specificity, immunogenicity, biodistribution and efficacy need to be overcome. Similarly, the concept of in utero gene therapy [58,60,61], which offers the opportunity for early clinical intervention at significantly reduced use of GMP-grade reagents, may provide an alternative solution to resource-demanding ex vivo HSC gene editing. For further progress here, additional data in large animal models and new legislations and guidelines need to be put in place first. Whatever the approach, willingness to pay for treatments needs to consider not only the annual and lifetime costs for alternative treatments but also indirect costs attributable to reduced productivity of patients and intangible costs associated with the pain and psychological suffering of the individual patient and their families. The societal cost considered and with ongoing developments further reducing cost of development and treatment, therapy by gene editing of HSC might become affordable and justified for an increasing proportion of chronic patients in the coming years.

## 5. Conclusions

Tremendous advances in isolation, characterization and manipulation of HSCs in combination with an increasingly versatile portfolio of gene editing tools have provided new insights into the underlying molecular mechanisms of diseases and established new frontiers for the treatment of many disorders, raising hope for the treatment of numerous as yet incurable diseases. HSCs in combination with gene editing tools have already shown success in preclinical treatments for a large number of hematological and non-hematological disorders. Partly owing to the relatively short history of programmable nucleases and of their application in vivo and in the clinic, many unknowns surrounding short- and long-term side effects of gene-edited products in patients remain. Where possible, DSB-free technology, short reagent exposure, editing of LTR-HSCs and safe cell handling, conditioning or in vivo treatment protocols need to be considered to reduce the potential risk to patients. Correspondingly, transparency and standardization of GMP manufacturing, preclinical safety and efficacy assessments and clinical follow-up are required, as are clinical trials with larger cohorts of patients and longer follow-up to ascertain the long-term efficacy and safety of these interventions. At the current trajectory of expansion and improvements for therapies by HSC-based gene editing, the technology could soon become a standard for the therapy of many life-threatening disorders, which in turn calls for standardized international regulatory frameworks. Realization of the promise of autologous, HSC-based gene editing therapies will depend on the scalability of clinical-grade reagents and procedures, and on the affordability and accessibility of these new treatments to the patients and communities in need.

## Figures and Tables

**Figure 1 cells-10-01492-f001:**
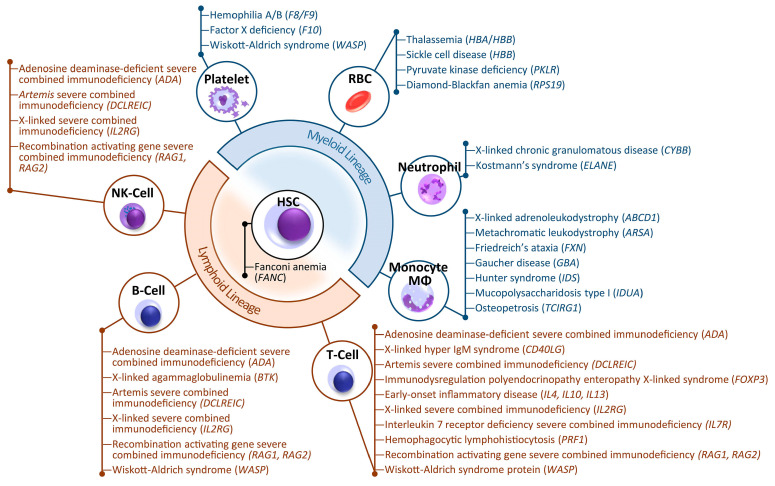
Overview of exemplary hereditary disorders potentially curable by editing of HSCs. Genes associated with each disease are indicated in parentheses. The cell types given indicate where protein expression or phenotypic correction are most apparent. Note that combined immunodeficiencies affect B cell function even when presenting with a B^+^ phenotype. Monocytes and macrophages (MΦ) may also act on cells and for disease correction outside the hematopoietic system (not shown). RBC—red blood cell, NK cell—natural killer cell.

**Figure 2 cells-10-01492-f002:**
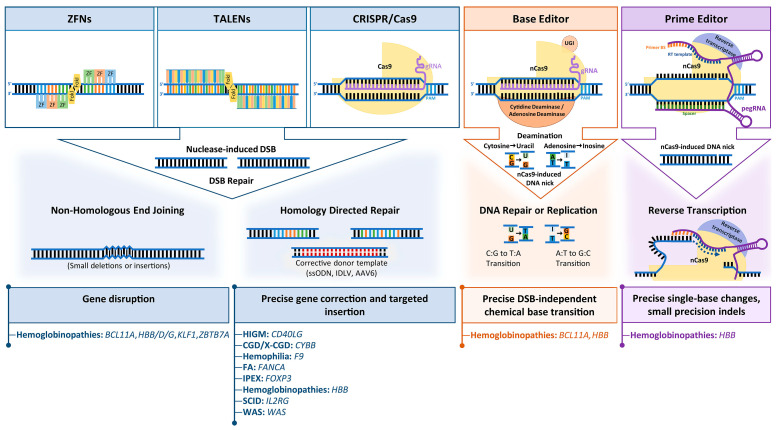
Structure, function and HSC-based application of common gene-editing platforms. ZFNs, TALENs and CRISPR/Cas9 are chimeric proteins comprising customizable sequence-specific DNA binding domain (e.g., zinc finger—ZF, transcription activator-like effector proteins—TALEs or single guide RNA—sgRNA) and a nonspecific nuclease that mediates DNA cleavage (e.g., FokI nuclease in the context of ZFNs and TALENs and Cas9 in the case of CRISPR/Cas9). DNA double-strand breaks generated by ZFNs, TALENs and CRISPR/Cas9 are mainly repaired via two endogenous pathways: (1) error-prone non-homologous end joining (NHEJ), which occurs throughout the cell cycle and corrects breaks through ligation of DNA ends, or (2) by precise homology-directed repair (HDR), in the presence of donor template provided, e.g., as synthetic single-stranded oligodeoxynucleotides (ssODNs), insertion-defective lentiviral vector (IDLV) or adeno-associated virus 6 vector (AAV6) components,. Base editors are chimeric proteins composed of a mutated nuclease, such as Cas9 nickase (nCas9), a catalytic domain capable of deaminating a cytidine or adenine base to induce transition mutations, and a uracil glycosylase inhibitor to prevent base excision repair of the transition event. Prime editors are chimeric proteins exploiting an extended gRNA, termed prime editing guide RNA (pegRNA), and a nCas9 fused to a reverse transcriptase, which nick the DNA to allow pegRNA binding of flanking gRNA to serve as primer of pegRNA-directed reverse transcription of the desired sequence change.

**Figure 3 cells-10-01492-f003:**
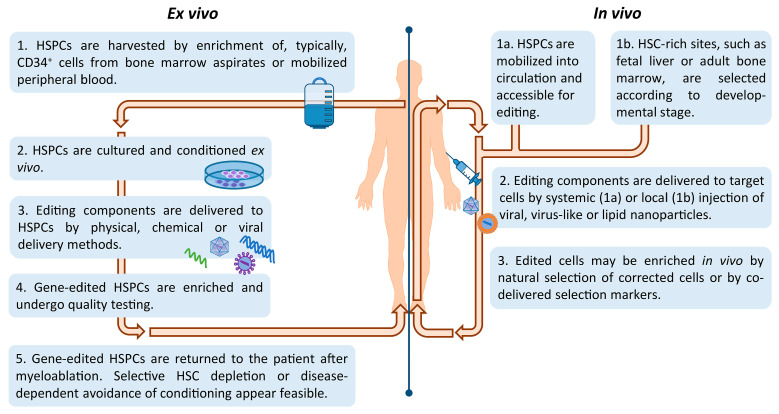
Ex vivo vs. in vivo HSC gene-editing. Steps shown for Ex vivo editing are widely applied for gene editing in in vivo animal models and now also in clinical trials for gene editing. Findings for therapy by gene addition indicate that gene editing, too, might benefit from selective HSC depletion by delivery of antibody-drug conjugates [55] and for suitable disorders, such as FA, from engraftment of corrected cells without conditioning [14]. Steps shown for In vivo editing are in part extrapolated for HSC-targeted approaches of gene addition [56,57,58] and in part based on the latest developments and concepts in the delivery of gene editing components and mRNAs [59,60,61,62].

**Table 1 cells-10-01492-t001:** Key features of gene editing tools.

	ZFN	TALEN	CRISPR
Target sequence	9–18 bp per ZFN monomer	14–20 bp per TALEN monomer	20 bp guide sequence plus PAM sequence
Recognition site	Zinc-finger protein	TALE protein RVD tandem repeat region of	Single-stranded single guide RNA (sgRNA)
Mode of recognition	Protein:DNA ZF modules; interference of neighboring recognition modules	Protein:DNA TALE RVDs; clear 2:1 amino acid:nucleotide code	RNA:DNA; 1:1 Watson–Crick base pairing
Endonuclease	FokI	FokI	Cas
Size (kb)	~1	~3	~3.5–4.5
Ease of engineering	Complicated—Requirement of substantial protein engineering	Simplified—Requirement of complex molecular cloning procedures	Simplest—Use of 20 nucleotide sgRNA sequence per target site
Off-target effects	High	Low	Variable
Size (kb)	~1	~3	~3.5–4.5
Ease of engineering	Complicated—Requirement of substantial protein engineering	Simplified—Requirement of complex molecular cloning procedures	Simplest—Use of 20 nucleotide sgRNA sequence per target site
Off-target effects	High	Low	Variable
Cost	High	Moderate	Low

Abbreviations: ZFN—Zing finger nucleases; TALEN—Transcription activator-like effector nucleases; CRISPR—Clustered regularly interspaced short palindromic repeats.

**Table 2 cells-10-01492-t002:** Overview of diseases addressed by gene editing in human primary hematopoietic cells.

Target Modification	Cell Source/Cell Type	Gene (Targeted Modification)	Gene Editing Strategy	Nuclease Delivery	Repair Donor Template	Efficiency of Gene Editing	Reference
**β-Hemoglobinopathies**							
Gene addition/correction	iPSCs	*HBB* (c.20 A>T)	ZFN	pDNA	pDNA	50% HDR	[125]
	iPSCs	*HBB* (c.20 A>T)	ZFN	pDNA	pDNA	38% HDR	[126]
	iPSCs	*HBB* (c.316-197C>T, c.126_129delCTTT)	TALEN	pDNA	pDNA	68% HDR	[127]
	iPSCs	*HBB*: c.-78A>G, c.126_129delCTTT	CRISPR/Cas9	pDNA	PiggyBac	23% HDR	[128]
	HSCs	*HBB* (c.20 A>T)	ZFN	mRNA	IDLV/ssODN	40% HDR	[82]
	iPSCs	*HBB* (c.52A>T)	CRISPR/Cas9	pDNA	dsDNA	17% HDR	[129]
	iPSCs	*HBB* (c.316-197C>T)	CRISPR/Cas9 & TALEN	pDNA	PiggyBac	12 & 33% HDR	[130]
	iPSCs	*HBB* (c.20 A>T)	CRISPR/Cas9	pDNA	dsDNA	40% HDR	[131]
	Human Embryos	*HBB* (c.126_129delCTTT)	CRISPR/Cas9	mRNA	ssODNs	14% HDR	[107]
	HSPCs	*HBB* (Exon1 & c.20 A>T)	CRISPR/Cas9	mRNA/RNP	AAV6	10% HDR	[106]
	iPSCs	*HBB* (c.126_129delCTTT)	CRISPR/Cas9	pDNA	ssODNs	5% HDR	[132]
	iPSCs	*HBB* (c.126_129delCTTT)	CRISPR/Cas9	pDNA	dsDNA	57% HDR	[133]
	HSPCs	*HBB* (c.20 A>T)	CRISPR/Cas9	mRNA	IDLV	20% HDR	[134]
	HSPCs	*HBB* (c.20 A>T)	CRISPR/Cas9	RNP	ssODNs	33% HDR	[81]
	iPSCs	*HBB* (c.126_129delCTTT)	CRISPR/Cas9	pDNA	dsDNA	NA	[135]
	Human Embryos	*HBB* (c.126_129delCTTT)	CRISPR/Cas9	RNP	ssODNs	25% HDR	[136]
	HSPCs	*HBB* (c.20 A>T)	CRISPR/Cas9	mRNA/RNP	ssODNs	9% HDR	[137]
	HSPCs	*HBB* (c.126_129delCTTT)	CRISPR/Cas9	pDNA	ssODNs	54% HDR	[138]
	iPSCs	*HBB* (Exon1, 3′ UTR, c.52A>T, c316-197C>T & c.126_129delCTTT)	CRISPR/Cas9	pDNA	pDNA	NA	[139]
	Human Embryos	*HBB* (c.-28A>G)	Base Editor	mRNA	-	23% BE	[140]
	HSPCs	*HBB* (c.93-21G>A)	CRISPR/Cas9, TALENs & ZFN	mRNA	ssODNs	8% HDR	[141]
	HSPCs	*HBB* (c.20 A>T)	CRISPR/Cas9	RNP	AAV6	70% HDR	[142]
Gene disruption	HSPCs	*B**CL11A* (*GATA 1*)	CRISPR/Cas9	LV	-	NA	[143]
	HSPCs	*HBG1/2* promoter	CRISPR/Cas9	LV	-	77% NHEJ	[144]
	HSPCs	*HBD*-*HBB*	CRISPR/Cas9 (SaCas9)	pDNA	-	31% NHEJ	[145]
	HSPCs	*BCL11A* (Exon 2)	CRISPR/Cas9	pDNA/mRNA	-	13% NHEJ	[146]
	HSPCs	*BCL11A* (Exon 2, GATAA)	ZFN	mRNA	-	45–50% NHEJ	[147]
	HSPCs	*HBA* MCS-R2 enhancer	CRISPR/Cas9	dsDNA	-	60% NHEJ	[148]
	HSPCs	*HBG*-*HBD*, *HBD*-*HBB*	CRISPR/Cas9	pDNA	-	20% NHEJ	[149]
	HSPCs	*HBG1/2* promoter	CRISPR/Cas9	ADV	-	24% NHEJ	[150]
	HSPCs	*BCL11A* (Exon 2)	ZFN	mRNA	-	72% NHEJ	[151]
	HSPCs	*HBB* (c.93-21G>A)	CRISPR/Cas9 & TALEN	RNP and mRNA	-	90% NHEJ	[152]
	HSPCs	*BCL11A* (*GATA 1*)	CRISPR/Cas9	RNP	-	87% NHEJ	[153]
	HSPCs	*HBG1/2*	TALEN	mRNA	-	74% NHEJ	[154]
**Severe Combined Immunodeficiencies**							
Gene addition/correction	T-cells	*IL2RG* (exon 5)	ZFN	pDNA	pDNA	7% HDR	[155]
	HSPCs	*IL2RG*	ZFN	IDLV	IDLV	39% HDR	[156]
	HSPCs	*IL2RG*	ZNF	mRNA	IDLV	6% HDR	[43]
	iPSCs	*JAK3* (c.1837C>T)	CRISPR/Cas9	pDNA	pDNA	73% HDR	[157]
	HSPCs	*IL2RG* (c.691G>A)	CRISPR/Cas9 & ZNF	mRNA	AAV6	27% HDR in CD34^+^CD133^+^CD90^+^	[98]
	T-cells	*IL2RG* (c.800delA/c.530A>G)	CRISPR/Cas9	RNP	ssDNA/dsDNA	25/22% HDR	[158]
	HSPCs	*IL2RG*	CRISPR/Cas9	RNP	AAV	45% HDR	[159]
**Wiskott–Aldrich Syndrome**							
Gene addition/correction	iPSCs	*WAS*	ZNF	pDNA	pDNA	NA	[160]
	HSPCs	*WAS*	CRISPR/Cas9	RNP	AAV6	60% HDR	[161]
**Fabry Disease**							
Gene addition/correction	HSPCs	*GLA* (TI in α-globin 5′ UTR)	AAV6	RNP	AAV6	NS (16x G:LA expression)	[162]
**Hurler Syndrome**							
Gene addition/correction	HSPCs	*IDUA* (TI in α-globin 5′ UTR)	AAV6	RNP	AAV6	NS (171x IDUA expression)	[162]
**Wolman Disease**							
Gene addition/correction	HSPCs	*LAL* (TI in α-globin 5′ UTR)	AAV6	RNP	AAV6	0.7 TI *LAL* copies/cell	[162]
**Chronic Granulomatous Disease**							
Gene addition/correction	iPSCs	*AAVS1*	TALENs	pDNA	pDNA	50% HDR	[163]
	iPSCs	*CYBB* (Int. 1 T>G)	ZFN	mRNA	AAV6	57% HDR	[164]
	iPSCs	*AAVS1*	ZFN	mRNA	pDNA	80% HDR	[165]
	HSPCs	*AAVS1*	CRISPR/Cas9	LV	AAV6	67% HDR	[97]
	iPSCs	*CYBB*	CRISPR/Cas9	pDNA	pDNA	17% HDR	[166]
	HSPCs	*CYBB* (C676T)	CRISPR/Cas9	mRNA	ssODN	21% HDR	[167]
	iPSCs	*NCF1B, NCF1C*	CRISPR/Cas9	mRNA	pDNA/rAAV2	90% HDR	[168]
	iPSCs	*NCF1*	CRISPR/Cas9	pDNA	pDNA	43–47%	[169]
	HSPCs	*CYBB*	CRISPR/Cas9	mRNA	ssODN	80%	[89]
**Immunodysregulation Polyendocrinopathy Enteropathy X-Linked**							
Gene addition/correction	HSPCs	*FOXP3*	CRISPR/Cas9	RNP	rAAV6	29% HDR	[170]
**Hyper IgM Syndrome**							
Gene addition/correction	T-cells	*CD40L*	TALEN	mRNA	rAAV	36–47%	[171]
	T-cells, CD34+	*CD40L* cDNA	TALEN & CRISPR/Cas9	mRNA	IDLV/AAV6	31- 34%	[99]
**Fanconi Anemia**							
Gene addition/correction	iPSCs	*FANCA*	ZFN	ADV	IDLV	40% HDR	[172]
	HSPCs	*FANCD1* (Exon 8)	CRISPR/Cas9	Plasmid	ssDNA	NA	[173]
	HSPCs	*FANCA*	ZFN	mRNA	IDLV	14% HDR	[174]
	HSPCs	(c.3558insG, c.295C>T), *FANCB*, *FANCC* (c.67delC), *FANCD1/BRACA2* (c.1596delA), *FANCD2* (c.718delT)	CRISPR/Cas9	pDNA	-	NHEJ	[175]
**Hemophilia A**							
Gene addition/correction	iPSC	*F8* (Inv 1)	TALENs	pDNA	-	Inversion 1%	[176]
	iPSC	*F8* (Inv 1 & 22)	CRISPR/Cas9	pDNA	-	Inversion 7%	[177]
	iPSC	*F8* (Inv 22)	TALENs	pDNA	pDNA	63% HDR	[178]
	iPSC	*F8*	CRISPR/Cas9	RNP	pDNA	66% HDR	[179]
**Hemophilia B**							
Gene addition/correction	Germline cells	*F9* (exon 8)	CRISPR/Cas9	RNP	ssDNA	53% HDR	[180]
	*HSPC*	*F9* (TI in α-globin 5′ UTR)	CRISPR/Cas9	AAV6	RNP	1 TI F9 copy/cell	[162]
**Amegakaryocytic Thrombocytopenia**							
Gene correction	HSPCs	*MPL* (c.814T>C)	CRISPR/Cas9	RNP	ssODN	NA	[181]
**HIV AIDS**							
Gene disruption	Th cells	*CCR5*	TALEN	mRNA GMP electroporation	**-**	>60% cells, 40% biallelic	[182]
	HSPC	*CCR5*	CRISPR/Cas9	pDNA electroporation	**-**	27%	[183]
Gene silencing	T cells	*CCR5* & *CXCR4*	TALE epigenome modifier	mRNA electroporation	**-**	% CpG methylation (10–90% CCR5, 2–13% CXCR4)	[184]

Abbreviations: AAV—adeno-associated virus vectors; ADV—Adenovirus; HDR—homology-directed repair; IDLV—integrase-deficient; iPSCs—induced pluripotent stem cells; LCLs—lymphoblastic cell lines; LV—lentiviral vector; NA—Not Available; NS—not specified; NHEJ—non-homologous end joining; pDNA—plasmid DNA; RNP—ribonucleoprotein; ssDNA—Single-stranded DNA; ssODN—single-stranded donor oligonucleotides; TI—targeted integration.

**Table 3 cells-10-01492-t003:** Overview of HSC-based clinical studies using gene editing.

NCT Number	Strategy	Phase	Country	Target/Modification	Nuclease	Delivery	Industry Sponsorship/Sponsor
**β-Thalassemia**							
NCT03655678	Ex vivo	I/II	Germany (Canada, Europe)	Autologous CD34^+^ HSPCs modified at the enhancer of the BCL11A	CRISPR/Cas9	RNP electroporation	CRISPR Therapeutics (Vertex Pharmaceuticals Incorporated)
NCT03728322	Ex vivo	I		HBB gene correction in patient specific iHSCs	CRISPR/Cas9	Not Specified	Allife Medical Science and Technology Co., Ltd.
NCT03432364	Ex vivo	I/II	USA	CD34+ HSPCs	ZFN	mRNA	Sangamo Therapeutics
**Sickle Cell Disease**							
NCT03653247	Ex vivo	I/II	USA (USA, Europe)	Autologous CD34^+^ HSPCs modified at the BCL11A erythroid enhancer	ZFN	mRNA	Rioverativ, a Sanofi company
NCT03745287	Ex vivo	I/II	USA (USA, Europe)	Autologous CD34^+^ HSPCs modified at the BCL11A erythroid enhancer	CRISPR/Cas9	RNP	CRISPR Therapeutics (Vertex Pharmaceuticals Incorporated)
NCT04443907	Ex vivo	I/II	USA	Autologous CD34^+^ HSPCs modified at the BCL11A erythroid enhancer	CRISPR/Cas9	RNP	Novartis Pharmaceuticals (Novartis Pharmaceuticals & Intellia Therapeutics)
NCT04774536	Ex vivo	I/II	USA	Autologous CD34^+^ HSPCs	CRISPR/Cas9	RNP/ssODN electroporation	University of California (Los Angeles), University of California (Berkeley)
NCT04853576	Ex vivo	I/II	USA	HBG1/2 promoter	CRISPR/Cas12	RNP	Editas Medicine
**Mucopolysaccharidosis I**							
NCT02702115	In vivo	I/II	USA	Insertion of corrected copy of α-L-iduronidase gene into the Albumin locus	ZFN	AAV	Sangamo Therapeutics
**Mucopolysaccharidosis II**							
NCT03041324	In vivo	I/II	USA	Insertion of corrected copy of α-L-iduronidase gene into the Albumin locus	ZFN	AAV	Sangamo Therapeutics
**Hemophilia B**							
NCT02695160	In vivo	I	USA, Europe	Insertion of corrected copy of the factor 9 gene into the Albumin locus	ZFN	AAV	Sangamo Therapeutics
**HIV AIDS**							
NCT02500849	Ex vivo	I	USA	Disruption in CD34^+^ HSPCs of CD4 co-receptor gene, CCR5	ZFN	mRNA electroporation	City of Hope Medical Center, Sangamo Therapeutics, California Institute for Regenerative Medicine
NCT03164135	Ex vivo	-	China	Disruption in CD34^+^ HSPCs of CD4 co-receptor gene, CCR5	CRISPR/Cas9	RNP	Academy of Military Medical Sciences, Peking University, Capital Medical University

Abbreviations: AAV—adeno-associated virus; RNP—ribonucleoprotein; ZFN—zinc finger nuclease.

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
