# Peer review of "Therapy Development by Genome Editing of Hematopoietic Stem Cells"

_cells, 2021, doi:10.3390/cells10061492_

Round 1

Reviewer 1 Report

The authors present a very comprehensive review on genome editing of hematopoietic stem cells. The review is very well written and covers a large span of issues.

The review is well balanced but I would add one aspect for ex vivo editing. The current protocols applied are based on toxic alkylating agents given before transfer of cells which in some patients are too toxic and if applied come along with an increased risk for hematopoietic malignancies like leukemias and MDS independent of risks derived from editing. Therefore, broader success depends critically on new methods to permit engraftment of ex vivo edited stem cells. 

Reviewer 2 Report

Koniali et al provide a thorough review of genomic editing in hematopoietic stem cells. The authors should be applauded for their attempt to be complete.  Indeed, the review is fairly complete, but as always when trying to be comprehensive, there are a few areas that warrant a bit more discussion.

  1. The limitations of gene editing in HSCs from a technical point of view have recently been reviewed (Morna et al, Klaver-Flores et al, Cannon et al, Thrasher et al). The clinical translation still faces significant hurdles, which is covered but not explicitly stated. It could be useful to refer to some of these reviews to keep the length of the article reasonable.
  2. The gene editing field is making quick progress also because of the progress made in gene additions, especially recent efforts using SIN-lentivirus for modification of autologous HSCs. This is a worthwhile point to be discussed.
  3. In fig 1 there are a few errors: X-SCID effects T cells (and NK) not B cells (at least in humans). Hence X-SCID should not be listed under B cells. ADA- SCXOID affects B, T and NK. It should not be only listed under B cells. Missing are the recombination deficiencies (RAG1, RAG2, Artemis) despite ongoing clinical trails for Artemis and RAG1-SCID.
